# Analyzing Gender Differences in Factors Affecting Depression among Multicultural Adolescents in South Korea: A Cross-Sectional Study

**DOI:** 10.3390/ijerph18073683

**Published:** 2021-04-01

**Authors:** Eun Jee Lee, Sookyung Jeong

**Affiliations:** 1Research Institute of Nursing Science, College of Nursing, Jeonbuk National University, Jeonju 54896, Korea; ejlee@jbnu.ac.kr; 2Department of Nursing, College of Medicine, Wonkwang University, Iksan 54538, Korea

**Keywords:** depression, gender difference, multicultural, adolescents, foreign mother

## Abstract

Suicide is the topmost cause of death among adolescents in South Korea and is deeply related to depression. This study aimed to identify gender differences in the factors affecting depression among multicultural adolescents. This study is a secondary analysis using data from a national survey, the Multicultural Adolescents Panel Study (MAPS) conducted in 2017. The participants were 1160 multicultural adolescents ranging from 15 to 18 years, living in Korea, and whose fathers were Koreans and mothers were foreigners. The results showed that depression scores were higher for females (18.35) than males (16.38, t = 6.42, *p* < 0.001). In total, seven factors affected depression among male multicultural adolescents’ and the model explained 50.5% of the total variance (F = 77.99, *p* < 0.001), while four factors affected female multicultural adolescents’ depression, and the model explained 51.4% of the total variance (F = 100.02, *p* < 0.001). Significant gender differences were found in factors that influence depression among multicultural adolescents. Therefore, depression prevention programs for multicultural adolescents need to vary according to gender. Additionally, these programs should target families and teachers of multicultural adolescents as well.

## 1. Introduction

The number of married migrant women has sharply increased in South Korea since the early 2000s, and this number has doubled over the last decade (2010–2019) [1]. Additionally, in 2019, students from multicultural backgrounds made up 2.5% of the total number of students in South Korea, which is 3.5 times more than that in 2012 [2]. As a result, the South Korean society is undergoing a change due to the sudden increase in multicultural families.

Adolescents experience physical, emotional, and social changes and growth [3]. Furthermore, middle and late adolescents experience impulsive thinking and unstable psychological conditions such as conflict, tension, and stress, due to different social roles and expectations from their parents and school [4]. This phenomenon is more prominent among adolescents belonging to multicultural families. A literature review reported that 115 Korean adolescents and 795 of those from multicultural families were more likely to engage in delinquency, violence, and game addictions; furthermore, it was revealed that 3995 Korean multicultural adolescents had a higher tendency to engage in substance abuse than monocultural adolescents in South Korea [5]. Additionally, 881 Mexican adolescents from multicultural families’ experienced higher stress levels, due to internal conflicts over bicultural adjustment and having parents belonging to different nationalities and cultures [6].

Few studies reported variables that positively influenced multicultural adolescents’ depression. They primarily emphasized the buffering role of family support and positive parenting style. These variables play an important role in increasing the self-esteem of multicultural adolescents, thereby decreasing their depression and suicidal ideation [7,8,9]. Moreover, multicultural adolescents’ mental health outcomes differed, depending on the experienced parenting style, i.e., monitoring had a higher positive effect on depression than neglect [10]. A previous report suggested that negative parenting styles led to loneliness and lack of close relationships with friends among multicultural adolescents. Subsequently, it caused social withdrawal and depression [11,12].

Another study emphasized that peer support, including friendship, is as important as family support in terms of adolescent depression [13]. During adolescence, as opposed to other age groups, friendship plays a valuable role, due to the major time spent with and high dependence on friends for social support [14,15]. Additionally, positive friendship is connected with school satisfaction [16]. However, studies have shown that multicultural adolescents experienced difficulties in building friendships and school satisfaction, due to their experience of bullying in school, violence, and ethnic discrimination from their peer group [17,18,19]. 

Migrant women, married to someone in a foreign country, experience acculturative stress while adapting to a new society. Many studies have proved that adapting to a foreign culture can cause psychological issues such as depression, anxiety, and identity confusion among married migrant women [20,21]. Moreover, there is a big communication gap between such couples on account of cultural differences and way of living; thus, marital conflicts are common between these couples. Marital conflicts combined with the acculturative stress experienced by the mother can hinder the attachment between the mother and her child [4,22]. A study about migrant women reported that their children are more likely to experience inertia and depression, due to their higher anxiety levels [2]. Furthermore, there is a possibility that migrant women’s issues, such as communication difficulties, martial conflict, and acculturative stress influence their multicultural adolescent children’s emotional and mental health, and these issues can subsequently cause depression. 

South Korean statistics report that in 2019, 28.2% of middle and high school students suffered from depression; this percentage was higher among female adolescents than male adolescents [2]. In addition, suicide has been ranked as the topmost cause of death among adolescents since 2011. A study with 17,195 multicultural adolescents in South Korea showed that they are more prone to psychological symptoms such as depressed mood and suicidal ideation than monocultural adolescents [23]. Moreover, multicultural adolescents are highly vulnerable to developing depression due to cultural identity confusion, home and family environment, and adolescence. According to the existing literature, they belong to the high-risk depression group and differ in vulnerability to depression as per their gender. Recently, several researchers have focused on the multicultural issues of Korea, since it is rapidly developing into a multicultural society [24,25,26,27]. However, very few studies have explored the factors influencing depression among multicultural adolescents with mothers of a foreign nationality. Therefore, it is necessary to identify the factors that influence adolescent depression by gender, due to the higher vulnerability of female adolescents to depression and the presence of a foreign mother and Korean father in most multicultural families within the Korean society. Therefore, we analyzed the factors that influence depression among adolescents from multicultural families, according to their gender.

## 2. Materials & Methods

### 2.1. Study Setting

A cross-sectional design was used in this study; a secondary data analysis was conducted on the data obtained from the national survey for multicultural adolescents in South Korea to confirm the gender differences in depression.

### 2.2. Participants and Procedure 

We used data from the 7th Multicultural Adolescents Panel Study (MAPS) for this study. A total of 1251 adolescents participated in the survey study in 2017; however, prior to data analysis, we excluded 91 adolescents, who either had a Korean mother and a foreign father or a foreign mother and a foreign father. This sample was selected based on the assumption that 95% of multicultural adolescents in South Korea have a foreign mother and a Korean father. Eventually, we selected a sample of 1160 adolescents from 1000 schools in 16 cities. 

The MAPS surveyed mothers and their multicultural adolescent children, who were in the 4th-grade of elementary school (9 to 10 years old) and could understand and respond to the contents of the questionnaire. It is a longitudinal survey that began in 2011 and will end in 2025, when the participating adolescents turn 24. The 7th-year (2017) data were used in this study because it was the most recently published. 

The MAPS data was collected by a trained professional, who visited the adolescents’ home and conducted an interview with them. The questionnaire for multicultural adolescents was in Korean, while the mothers were provided with a questionnaire translated into nine languages (as per their countries of origin: China, China (Korean Chinese), Vietnam, Philippines, Japan, Thailand, and others) along with the Korean questionnaire. In the first stage, a stratified random sampling technique was applied. Followed by probabilistic sampling in the second stage, to extract schools; multicultural adolescents from the extracted schools were investigated in this study. The sample retention rate was 77.0%, and sample replacement was not performed for sample deviation.

### 2.3. Measures

#### 2.3.1. Depression

The revised version of the Symptom Checklist-90-Reversion [28] was used to detect depression symptoms [29]. It has 10 items with a 4-point Likert-type response scale. A higher score on this scale indicates a higher level of depression. In Shon’s and Park’s studies [10,30] with Korean multicultural adolescents, this scale reported a reliability of 0.91 and 0.90, respectively. In this study, this scale reported high internal consistency with a Cronbach’s alpha of 0.91.

#### 2.3.2. Self-Esteem

Rosenberg’s [31] scale was used to evaluate individual self-esteem in this study, but was modified to incorporate easy words and sentences, in accordance with the age of the respondents. It has nine items with a 5-point Likert-type response scale. A higher score on this scale indicates higher self-esteem. The reliability of the original Rosenberg Self-Esteem Scale was 0.92; in previous studies, which used the revised version of Rosenberg’s scale, the internal consistency of this tool ranged from 0.83 to 0.84 [10,24,30]. In this study, the scale reported high internal consistency with a Cronbach’s alpha of 0.87.

#### 2.3.3. Social Withdrawal

Social withdrawal was evaluated using the Behavior Problem Scale for Children and Adolescents developed by Kim and Kim [14] and revised by Lee at al. [29]. This scale assessed the degree of appropriate social relationships by evaluating the participants’ interactions with the surrounding people and the environment. It has five items with a 4-point Likert-type response scale. A higher score indicated a higher level of social withdrawal. The Cronbach’s alpha of this scale in Kim and Kim’s study was 0.79 [32], and it was 0.85, 0.84, and 0.86 in the 4th, 5th, and 6th, respectively [33]. In this study, the scale reported high internal consistency with a Cronbach’s alpha of 0.91.

#### 2.3.4. Attitudes toward Bicultural Adjustment

The Social, Attitude, Familial, and Environmental Acculturative Stress Scale was used to measure the degree of adaptation to the culture of South Korea and the foreign mother’s country, including music, movies, clothing, participation in cultural activities, future residence, and countries with schools they want to attend. The scale was developed by Hovey and King [34] and was translated and revised by Nho [35]. It has 10 items with a 4-point Likert-type response scale. Higher scores indicated higher interest in biculturalism. Hovey and King reported a reliability of 0.89 [34] and No’s study reported an alpha coefficient of 0.76 for his study [35]. In this study, the scale reported high internal consistency with a Cronbach’s alpha of 0.75.

#### 2.3.5. Mother’s Acculturation Stress

The Acculturative Stress Scale for International Students was used in this study; it was developed by Sandhu and Asrabadi [36] and revised and modified by Lee [37] and Lee [38], respectively. This scale is divided into three subscales: perceived discrimination, homesickness, and others. It has eight items with a 5-point Likert-type response scale. A higher score indicted a higher adjustment for biculturalism. Sandhu and Asrabadi reported alpha coefficients ranging from 0.87 to 0.95 [36]. Additionally, the Cronbach’s alpha reported in Lee’s [37] and Lee’s [38] studies were 0.93 and 0.94, respectively. In this study, the scale reported high internal consistency with a Cronbach’s alpha of 0.86.

#### 2.3.6. Parenting Style

##### Monitoring

The Parenting Behavior Inventory Scale [39] measures the degree to which parents know about the different aspects of their children’s life, such as school and peer relationships, was used in this study. It includes eight subscales: monitoring, reasoning, inconsistency, over-expectation, intrusiveness, physical abuse, neglect, and affection. Among these subscales, three subscales with three monitoring items were used in this study with a 4-point Likert type response scale. A higher score indicated parents higher interest in their children’s daily lives. The alpha coefficients of Parental Behavior Inventory Scale ranged from 0.77 to 0.79. In this study, the scale reported high internal consistency with a Cronbach’s alpha of 0.86.

##### Neglect

Neglect refers to the lack of parenting behavior necessary for children’s growth and development. Neglect was assessed using five items from the Heo’s Parenting Behavior Inventory Scale [39] and two items from Kim’s child abuse scale [40], which was revised by Lee [29]. In total, seven items with a 4-point Likert-type response scale were used in this study. Higher scores indicated higher neglect in parent’s behavior. Previous studies using this scale reported alpha coefficients ranging from 0.67 to 0.81 [41,42]. In this study, the scale reported high internal consistency with a Cronbach’s alpha of 0.83.

#### 2.3.7. Family Support

In this study, the family support subscale of Han’s [43] social supportive perception scale was used. In total, seven items, modified and revised by researchers, with a 4-point Likert type response scale were used in this study. A higher score indicated a higher level of family support. In Lee’s study, the internal consistency of this scale was 0.94 [44]. In this study, the scale reported high internal consistency with a Cronbach’s alpha of 0.95.

#### 2.3.8. Friendships

Few items about friendship were extracted from Hwang and Kim’s scale [45] and Kim’s scale [46]; these were revised and modified [45,46]. The final scale consisted of four items with a 5-point Likert-type response scale. The scale reliability reported by the Cronbach’s alpha was 0.80, in Kim’s study [46], while Nho’s study reported an internal consistency of 0.74 [47]. In this study, the scale reported high internal consistency with a Cronbach’s alpha of 0.91. 

#### 2.3.9. Satisfaction with School Life

One item from the Korean Education and Employment Panel survey for students [48] was used. It was “Are you satisfied with your current school life?” The scores ranged from 1 to 5.

#### 2.3.10. Health Condition 

It was scored by adolescents to measure the degree of subjective health perception. There are three categories: healthy, moderate, and unhealthy. The scores ranged from 1 to 3.

#### 2.3.11. Others—Demographic Characteristics

Information about the size of the respondent’s residential area, their economic status, parents’ marital status, and the age of the father and mother were collected as demographic characteristics.

### 2.4. Data Analysis

SPSS (Version 26.0) was used for data analysis. The demographics of multicultural adolescents were categorized by gender and analyzed using descriptive statistics (frequency, percentage, mean, and standard deviation). Skewness, kurtosis, and Levene’s test were assessed to verify the normality and homogeneity of variance assumptions. All continuous variables in this study satisfied the normality and homogeneity of variance assumptions. Gender differences in the demographics and variables among multicultural adolescents were analyzed using the Chi-square test and independent *t*-test. Additionally, one-way ANOVA was used to analyze the differences in depression levels according to the health condition and satisfaction with school life among male and female multicultural adolescents. Pearson’s correlation coefficients were used to analyze the relationship between male and female adolescents’ depression and other research variables. Factors influencing depression symptoms in participants were examined by performing multiple linear regression analyses. Furthermore, stepwise multiple linear regression analyses were performed to analyze the factors influencing depression symptoms in male and female multicultural adolescents. 

## 3. Results

### 3.1. Gender Differences in Demographics of Multicultural Adolescents and Their Parents 

We analyzed data from 1160 multicultural adolescents and parents in this study. The participants ranged from 15 to 18 years, and their mother and father were of foreign and Korean nationalities, respectively. The differences in participant demographics and variables are presented in Table 1. Female adolescents accounted for 51.4% of the total sample. Adolescents aged 16 made up 89.2% of the sample (M = 15.95 years). However, no significant gender differences were observed in the adolescents’ demographic characteristics. Depression, self-esteem, social withdrawal, attitudes toward bicultural adjustment, parenting style (monitoring), and family support showed significant gender differences. Moreover, depression, social withdrawal, attitudes toward bicultural adjustment, and parenting style (monitoring) in multicultural adolescents had higher reported scores among females than males, while self-esteem and family support scores were higher among males than among females. Additionally, the effect size of gender differences was 0.363 for depression (small-medium), 0.203 (small) for self-esteem, and 0.200 (small) for attitudes toward bicultural adjustment. The effect sizes of other variables were less than 0.2. Additionally, the mothers’ scores reported no significant gender differences in acculturation stress, parenting style (neglect), and friendships.

### 3.2. Gender Differences in Depression according to Multicultural Adolescents’ Health Condition and Satisfaction with School Life

As shown in Table 2, there was a statistically significant difference (F = 23.71, *p* < 0.001) in the scores for depression depending on the level of school life satisfaction among multicultural male adolescents; however, no such difference was observed depending on the health condition. Conversely, among female adolescents, the scores for depression were significantly higher (F = 9.74, *p* < 0.001), when perceived health condition was normal compared to when perceived health condition was good. Additionally, depression scores varied significantly (F = 46.01, *p* < 0.001) depending on the degree of school life satisfaction.

### 3.3. The Relationship among Research Variables by Gender of Multicultural Adolescents

Table 3 presents the relationship between the variables of male and female multicultural adolescents. Depression had a significant negative correlation with attitudes toward bicultural adjustment (r = −0.245, *p* < 0.001), self-esteem (r = −0.587, *p* < 0.001), monitoring parenting style (r = −0.341, *p* < 0.001), family support (r = −0.506, *p* < 0.001), and friendships (r = −0.472, *p* < 0.001), and a significant positive correlation with social withdrawal (r = 0.474, *p* < 0.001), mothers’ acculturation stress (r = 0.212, *p* < 0.001), neglect parenting style (r = 0.390, *p* < 0.001) among male adolescents. Furthermore, depression had a significant negative correlation with attitudes toward bicultural adjustment (r = −0.264, *p* < 0.001), self-esteem (r = −0.632, *p* < 0.001), monitoring parenting style (r = −0.312, *p* < 0.001), family support (r = −0.365, *p* < 0.001), and friendship (r = −0.373, *p* < 0.001), and a significant positive correlation with social withdrawal (r = 0.459, *p* < 0.001), mothers’ acculturation stress (r = 0.135, *p* < 0.001), and neglect parenting style (r = 0.380, *p* < 0.001).

### 3.4. Factors Influencing Multicultural Adolescents’ Depression by Gender

Table 4 presents the factors that influence multicultural adolescents’ depression categorized by gender. Individual factors (health condition, self-esteem, and social withdrawal), cultural factors (attitudes toward bicultural adjustment), family factors (mothers’ acculturation stress, parenting style, and family support), and school factors (satisfaction with school life and friendships) were analyzed as independent variables to understand their effect on depression. The variance inflation factor (VIF) was 1.03–1.72 and 1.12–1.53 for male and female multicultural adolescents, respectively; all variables had a VIF of less than 10 and showed no multicollinearity. The Durbin–Watson D statistics revealed coefficients of 2.048 and 2.019 for male and female multicultural adolescents, respectively, confirming the absence of a positive or negative autocorrelation.

The regression analysis model reported a total of ten factors (gender, unsatisfied or moderate satisfying school life, self-esteem, social withdrawal, mother’s acculturation stress, monitoring and neglect parenting style, family support, friendship) that influenced depression in all participants. The total explanatory power of the model was 51.1% (F = 90.70, *p* < 0.001). Gender had a small-medium effect size on depression (Cohen’s d = 0.302).

A total of seven factors were found to influence depression among male multicultural adolescents. Depression scores for males were higher when associated with lower self-esteem, higher levels of social withdrawal, low family support, poor friendships, higher levels of mothers’ acculturation stress, lower satisfaction with school life, and poor health conditions. The total explanatory power of the model was 50.5% (F = 77.99, *p* < 0.001). On the other hand, six factors were found to influence depression in female multicultural adolescents. Depression scores for females were higher when associated with low self-esteem, higher levels of social withdrawal, lower satisfaction with school life, higher prevalence of negligent parenting style, and perceived disinterest of parents. The total explanatory power of the model was 51.4% (F = 100.02, *p* < 0.001).

## 4. Discussion

This study analyzed secondary data from MAPS (2017) to identify the factors that influenced depression among male and female multicultural adolescents.

Gender had a small-medium effect size on depression in multicultural adolescents. The depression levels among female multicultural adolescents were higher than among male multicultural adolescents. These results are consistent with previous studies, which reported that women are more depressed than men throughout their lives [49,50,51]. These studies stated that women were more susceptible to depression than men because of their encounters with negative life events [52], coping styles [53,54], and gender differences in pubescent hormonal changes such as increase in estrogen [55]. Furthermore, since adolescence is an unstable period, where biological, cognitive, and social changes occur rapidly, adolescents are vulnerable to developing depression and suicidal symptoms [56]. In fact, suicide accounts for 37.5% of adolescent deaths in South Korea and has been the leading cause of death during the last decade [2]. Unfortunately, the rate is growing every year. Moreover, multicultural adolescents are more vulnerable to depression because they are prone to racial identity confusion, which enhances depressive symptoms [57]. Previous studies have shown that depression in adolescence is likely to transpire into depression during adulthood [49]; therefore, constant efforts are required to detect depression among adolescents at an early stage and tackle their symptoms.

The findings of the current study revealed that depression among male multicultural adolescents was affected by family support, friendships, health condition, and their mother’s acculturation stress. In Asian cultures, men are expected to lead their families, maintain family solidarity, and continue family traditions [58]. Thus, family support is perceived to be a valuable factor for male adolescents. Many research studies have shown that multicultural adolescents also need family support to overcome depression; they require strong family support to protect themselves and endure depression caused by social discrimination [57,59]. Relationships with friends were shown to be a significant influential factor for depression in male adolescents as well. During adolescence, friendship becomes increasingly important [60], and this relationship influences male adolescents more than female adolescents. Thus, maintaining a good rapport with friends is crucial to avoid depression among male adolescents [61,62,63]. Although relationships with friends can act as a major stressor during adolescence, it can also become an opportunity to learn social skills [64]. Therefore, male adolescents should be provided with the necessary professional counseling opportunities and be rid of their stress arising from relationships with friends. Additionally, the mother’s acculturation stress was another factor that influenced depression in male multicultural adolescents. In Nam and Lee’s study [23], mothers who were born in a foreign country experienced severe stress and family conflict because of cultural differences, which weakened their relationship with their child. Consequently, additional efforts are required from family members of such mothers to learn about their culture. Similarly, in addition to multicultural programs for mothers from foreign countries, various national policies and community efforts to manage their psychological and physical health need to be implemented.

On the other hand, depression among female multicultural adolescents in this study was influenced by parenting styles (monitoring and neglect). This finding implies that children with parents, who are least interested in their children, become more depressed. Female adolescents especially showed a decrease in problem behaviors and had a positive influence on their mental health when they felt close to their parents and regularly communicated with them [65]. However, parents in multicultural families showed lower levels of parenting satisfaction and efficacy due to acculturation stress and language barriers, causing a communication gap with each other; additionally, it rendered them unable to provide adequate support to their children [66,67]. In order to improve parental relationships for better mental health among female multicultural adolescents, it is essential to introduce various support programs providing parenting education to improve communication, introduction to the culture of mother’s country, and language education for parents. Subsequently, the findings of this study are consistent with several studies showing that the lack of parental support had a greater influence on depression than peer group support in female adolescents [68,69]. This suggests that female adolescents are more influenced by their parents’ emotional support and parenting style [61,70]. Additionally, female adolescents seek and want more emotional support [71], and they are more vulnerable to losing control of their emotions than male adolescents when experiencing difficulties in interpersonal relationships [70,72]. Therefore, depression in female adolescents should be treated with careful consideration of parental participation and parents’ emotional support.

In this study, the factors commonly affecting depression among male and female multicultural adolescents were self-esteem, social withdrawal, and school life satisfaction. Several other studies reported similar findings among adolescents for self-esteem [51,54,59], social support [54], and school life satisfaction [50,51].

Since this study is a secondary analysis, using the data obtained from the 2017 MAPS of the Korea Youth Policy Institute, there was a limitation to the analysis; the cause of the variable relationships could not be established, and we could not analyze additional variables from the data. Additionally, while the reliabilities of most scales in this study were indicated in previous studies, their validities were not evaluated. Therefore, future researchers should verify the validities of these instruments to clearly establish the relationship among these variables. However, despite these limitations, generalization is relatively possible, because this survey was conducted by applying random stratified sampling to multicultural adolescents across the country.

## 5. Conclusions

The findings of this study provide insight into the gender differences in depression among multicultural adolescents. It is essential to identify factors related to depression based on gender differences to detect the frequency and cause of vulnerability and prevalence of depression among adolescents. Gender differences existed based on family support, friendships, mothers’ acculturation stress, and parenting styles. Thus, healthcare providers should develop depression prevention programs not only for adolescents but also for parents and teachers. It is also necessary to vary these programs according to gender.

## Figures and Tables

**Table 1 ijerph-18-03683-t001:** Gender differences in the demographics of multicultural adolescents and their parents (n = 1160).

Variables	Categories	Total (n = 1160)	Male (n = 564, 48.6%)	Female (n = 596, 51.4%)	x^2^ or t	*p*	Cohen’s d
		n or Mean	n (%) or Mean (SD)	n (%) or Mean (SD)			
Size of the residential area	Large cities	283	147 (26.1)	136 (22.8)	2.35	0.309	
	Small and medium cities	513	250 (44.3)	263 (44.1)			
	Rural areas	364	167 (29.6)	197 (33.1)			
Economic status	Poor	596	275 (49.6)	321 (55.0)	3.36	0.186	
	Average	515	266 (48.0)	249 (42.6)			
	Rich	27	13 (2.4)	14 (2.4)			
Parents’ marital status	Married/Cohabiting	1082	530 (95.7)	552 (94.5)	0.80	0.371	0.052
	Divorced/Estranged/widowed	56	24 (4.3)	32 (5.5)			
Age of father			52.31 (4.32)	52.26 (4.62)	0.20	0.845	
Age of mother			46.72 (4.95)	46.54 (5.23)	0.57	0.569	
Father’s nationality	South Korea	1138	554 (48.7)	584 (31.3)	-	-	
Mother’s nationality	Chinese	81	38 (46.9)	43 (53.1)	3.42	0.754	
	Korean–Chinese	207	100 (48.3)	107 (51.7)			
	Vietnam	28	15 (53.6)	13 (46.4)			
	Philippines	296	151 (51.0)	145 (49.0)			
	Japan	424	202 (47.6)	222 (52.4)			
	Thailand	46	18 (39.1)	28 (60.9)			
	ETC	56	30 (53.6)	26 (46.4)			
Depression		17.33	16.38 (5.22)	18.35 (5.61)	6.42	<0.001	0.363
Self-esteem		34.33	34.93 (5.66)	33.77 (5.77)	3.45	0.001	0.203
Social withdrawal		11.94	11.64 (3.62)	12.22 (3.60)	2.74	0.006	0.161
Attitudes toward bicultural adjustment		29.08	28.67 (3.79)	29.46 (4.08)	3.57	<0.001	0.200
Mother’s acculturation stress		19.22	19.21 (5.67)	19.22 (5.61)	0.05	0.960	0.002
Parenting style—monitoring		9.56	9.45 (1.77)	9.67 (1.76)	2.18	0.030	0.125
Parenting style—neglect		12.13	12.14 (3.60)	12.13 (3.49)	0.03	0.978	0.003
Family support		22.15	22.42 (3.75)	21.90 (3.85)	2.36	0.018	0.137
Friendships		16.05	15.89 (2.83)	16.12 (2.91)	1.41	0.158	0.080

**Table 2 ijerph-18-03683-t002:** Gender differences in depression according to multicultural adolescents’ health condition and school life satisfaction (n = 1160).

Variables		Male (n = 564)							Female (n = 596)						
		N	%	Mean	±	SD	t or F	*p* (Scheffé)	N	%	Mean	±	SD	t or F	*p* (Scheffé)
Health condition	Total	554	100.0	16.32	±	5.22	0.34	0.713	584	100.0	18.26	±	5.64	9.74	<0.001
	Bad ^a^	54	9.7	16.72	±	5.25			67	11.5	19.42	±	5.93		a,b > c
	Moderate ^b^	253	45.7	16.14	±	5.26			269	46.1	19.05	±	5.83		
	Good ^c^	247	44.6	16.40	±	5.17			248	42.4	17.08	±	5.14		
Satisfaction with school life	Total	564	100.0	16.31	±	5.20	23.71	<0.001	596	100.0	18.30	±	5.66	46.01	<0.001
	Unsatisfied ^a^	32	5.7	20.44	±	5.78		a > b > c	47	7.9	23.83	±	5.89		a > b > c
	Moderate ^b^	149	26.4	17.77	±	5.66			190	31.9	19.68	±	5.32		
	Satisfied ^c^	383	67.9	15.39	±	4.65			359	60.2	16.84	±	5.17		

^a,b,c^; post-hoc test.

**Table 3 ijerph-18-03683-t003:** The relationship among research variables according to gender of multicultural adolescents.

Variables	(1) Depression		(2)		(3)		(4)		(5)		(6)		(7)		(8)	
Male																
(2) Attitudes toward bicultural adjustment	−0.245	**														
(3) Self-esteem	−0.587	**	0.375	**												
(4) Social withdrawal	0.474	**	−0.065		−0.346	**										
(5) Mother’s acculturation stress	0.212	**	−0.040		−0.180	**	0.077									
(6) Parenting style—monitoring	−0.341	**	0.273	**	0.398	**	−0.182	**	−0.086	*						
(7) Parenting style—neglect	0.390	**	−0.324	**	−0.510	**	0.172	**	0.078		−0.384	**				
(8) Family support	−0.506	**	0.410	**	0.531	**	−0.251	**	−0.108	*	0.494	**	−0.540	**		
(9) Friendships	−0.472	**	0.322	**	0.504	**	−0.279	**	−0.174	**	0.285	**	−0.337	**	0.430	**
Female																
(2) Attitudes toward bicultural adjustment	−0.264	**														
(3) Self-esteem	−0.632	**	0.344	**												
(4) Social withdrawal	0.459	**	−0.048		−0.352	**										
(5) Mother’s acculturation stress	0.135	**	−0.057		−0.152	**	0.118	**								
(6) Parenting style—monitoring	−0.312	**	0.385	**	0.332	**	−0.103	*	−0.032							
(7) Parenting style—neglect	0.380	**	−0.333	**	−0.427	**	0.107	**	0.030		−0.455	**				
(8) Family support	−0.365	**	0.481	**	0.439	**	−0.137	**	−0.081		0.475	**	−0.528	**		
(9) Friendships	−0.373	**	0.312	**	0.439	**	−0.228	**	−0.149	**	0.233	**	−0.306	**	0.363	**

** *p* < 0.01; * *p* < 0.05, two tailed.

**Table 4 ijerph-18-03683-t004:** Factors influencing multicultural adolescents’ depression by gender (n = 1160).

Gender	Variables	Adjusted R^2^	B	β	t	*p*
All participants (n = 1160) Enter method	(Constants)		26.02		13.91	<0.001
	Gender: female vs. male (R)		1.23	0.11	5.14	<0.001 Cohen’s d = 0.302
	Health condition: bad vs. good (R)		0.11	0.01	0.28	0.784
	Health condition: moderate vs. good (R)		−0.15	−0.01	−0.58	0.562
	School life: unsatisfied vs. satisfied (R)		2.67	0.12	5.40	<0.001
	School life: moderate vs. satisfied (R)		0.62	0.05	2.19	0.029
	Self-esteem		−0.32	−0.33	−11.80	<0.001
	Social withdrawal		0.43	0.28	12.36	<0.001
	Attitude toward bicultural adjustment		0.02	0.01	0.46	0.648
	Mother’s acculturation stress		0.05	0.05	2.25	0.025
	Parenting style (monitoring)		−0.17	−0.05	−2.17	0.030
	Parenting style (neglect)		0.11	0.07	2.74	0.006
	Family support		−0.12	−0.09	−2.97	0.003
	Friendship		−0.14	−0.07	−2.82	0.005
	Adjusted R^2^ = 0.511, F = 90.70, *p* < 0.001					
Male (n = 564) Stepwise method	(Constants)		29.37		16.70	<0.001
	Self-esteem	0.341	−0.25	−0.27	−6.86	<0.001
	Social withdrawal	0.426	0.41	0.28	8.51	<0.001
	Family support	0.467	−0.28	−0.20	−5.36	<0.001
	Friendships	0.484	−0.27	−0.15	−3.92	<0.001
	Mothers’ acculturation stress	0.491	0.09	0.10	3.06	0.002
	School life: unsatisfied vs. satisfied (R)	0.495	1.46	0.07	2.11	0.035
	Health condition: moderate vs. good (R)	0.498	−0.68	−0.07	−2.11	0.036
	Adjusted R^2^ = 0.498, F = 77.99, *p* < 0.001					
Female (n = 596) Stepwise method	(Constants)		25.43		11.89	<0.001
	Self-esteem	0.396	−0.39	−0.41	−11.19	<0.001
	Social withdrawal	0.468	0.46	0.30	9.44	<0.001
	School life: unsatisfied vs. satisfied (R)	0.486	3.39	0.16	5.13	<0.001
	Parenting style (neglect)	0.502	0.18	0.11	3.20	0.001
	School life: unsatisfied vs. satisfied (R)	0.505	0.81	0.07	2.16	0.031
	Parenting style (monitoring)	0.509	−0.23	−0.07	−2.15	0.032
	Adjusted R^2^ = 0.509, F = 100.02, *p* < 0.001					

R^2^ = R squared or coefficient of determination; B = nonstandardized coefficient; β = standardized coefficient; (R) = Reference.

## Data Availability

The following are available online at https://www.nypi.re.kr/archive/mps/program/examinDataCode/dataDwloadAgreeView?menuId=MENU00226, accessed on 21 February 2020. The data can be downloaded after permission is granted by the National Youth Policy Institute.

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
