# Peer review of "Analyzing Gender Differences in Factors Affecting Depression among Multicultural Adolescents in South Korea: A Cross-Sectional Study"

_ijerph, 2021, doi:10.3390/ijerph18073683_

Round 1
Reviewer 1 Report
I appreciate the opportunity to review this study. The authors have a lot of information and have managed to create a very interesting and well-structured manuscript. However, I add a few comments.
- Add whether it is a cross-sectional or longitudinal study.
- Replace "x = x" with "x=x".
- Include information on methodology, no information on whether you perform regression analysis, multifactorial, etc.
INTRODUCTION
- Your study is based on the problems presented by multicultural families. Indicate the number of participants and country of study 5, 6 and 10.
- Although it includes many variables, it is not clear the relationship between all the variables included in the study and the relationship with depression due to being an adolescent of multicultural origin.
METHOD
- Add information about the place where the data were collected. was it in the adolescents' home? was it at school?
- Is this study approved by any ethics committee?
- You have to add information about the validity and reliability of the different scales reported in previous studies.
- Why being a longitudinal study use cross-sectional data?
RESULTS
- Replace "x = x" with "x=x".
- Although the results are well presented, we know that the difference in means being statistically significant is not enough. Therefore, you need to add a statistic to measure the effect size of the means. In that case, you will need to indicate which mean differences have a small, medium or high size. In that case, you could look at the significance of the score difference in the variables with significance between male and female scores (depression, self-esteem, social withdrawal and Attitudes toward bicultural adjustment). This is necessary because your work focuses on the differences in scores between men and women.
- Add space so that the word "depression" appears in full in Table 3.
- In Table 4 it is not necessary to add the column "R2" because "Adjusted R2" already appears.
DISCUSSION
- When you include information on the effect size of the difference in means between males and females, your discussion will make more sense. Currently, you cannot claim to highlight the difference in scores between men and women.
- One of your questionnaires has a cronback alpha of less than 0.8. That may be a limitation, as well as, that cross-sectional data were used. However, it only indicates a limitation.
CONCLUSION
- You cannot say that you have to vary programs by gender without using an effect size to see the impact of differences.
Author Response
Dear Editors and Reviewer,
We are grateful to the editor and reviewer for the thoughtful review and edits of our manuscript entitled “ Analyzing gender differences in factors
affecting depression among multicultural adolescents in South Korea (# ijerph-1125734)”. Thank you for the opportunity to make recommended
improvements, technically and conceptually in order to further develop this manuscript. We appreciate your time, knowledge, and attention to
detail. Please see attached the resubmitted manuscript.
Drs. Lee and Jeong

Reviewer 2 Report
An interesting idea for an article, well thought out and supported by literature on the subject. The article is logical and reads well, and may complement what is already known. The simple and clear layout of the article makes reading easier. Despite my good opinion of the article, I wanted to make a few minor comments:
1. Using independent t-test requires fulfilling several assumptions. In your text there is no information about normality of the distribution of variables or homogeneity of variances. One can doubt if appropriate test was chosen.
2. Similar problem is with using Pearson’s correlation coefficients. I deeply doubt if it was a proper choice.
Could you comment on correlation results? You establish statistically significant correlation between all but three variables in male, and all but four variables in female.
3. Readability of tables is not better when you use description “ n or Mean” or “ % or SD”.
4. In recent years quite impressive number of researchers worked on multicultural adolescents in South Korea. In my opinion is a must to make clear what your work add to previous researches. You should include and discuss with articles of
- Joung KH and Chung SS Factors Related to Depressive Symptoms Among Multicultural Adolescents in Korea 2020,
- Song MK, Yoon JY, Kim E. Trajectories of Depressive Symptoms among Multicultural Adolescents in Korea: Longitudinal Analysis Using Latent Class Growth Model. Int J Environ Res Public Health. 2020,
- Park GR, Son I, Kim SS. Perceived Ethnic Discrimination and Depressive Symptoms Among Biethnic Adolescents in South Korea. 2016,
- Jang J, Park EC, Lee SA, et al. Association between Parents' Country of Birth and Adolescent Depressive Symptoms: the Early Stages of Multicultural Society. 2018,
- Park S, Lee Y. Factors that Affect Suicide Attempts of Adolescents in Multicultural Families in Korea. Int J Environ Res Public Health. 2016,
Author Response

(The authors gave the same response as above.)
